# Stromal Factors as a Target for Immunotherapy in Melanoma and Non-Melanoma Skin Cancers

**DOI:** 10.3390/ijms23074044

**Published:** 2022-04-06

**Authors:** Taku Fujimura

**Affiliations:** Department of Dermatology, Tohoku University Graduate School of Medicine, 1-1 Seiryo-machi, Aoba-ku, Sendai 980-8574, Japan; tfujimura1@mac.com; Tel.: +81-22-717-7271

**Keywords:** cancer stroma, TAMs, CAF, chemokines, angiogenetic factors

## Abstract

Immune checkpoint inhibitors (ICIs), such as anti-programmed cell death 1 (PD1) antibodies (Abs) and anti-cytotoxic T-lymphocyte associated protein 4 (CTLA4) Abs, have been widely administered for not only advanced melanoma, but also various non-melanoma skin cancers. Since profiles of tumor-infiltrating leukocytes (TILs) play important roles in immunotherapy using ICIs, it is important to evaluate cancer stromal cells such as tumor-associated macrophages (TAMs) and cancer-associated fibroblasts (CAFs), as well as stromal extracellular matrix protein, to predict the efficacy of ICIs. This review article focuses particularly on TAMs and related factors. Among TILs, TAMs and their related factors could be the optimal biomarkers for immunotherapy such as anti-PD1 Ab therapy. According to the studies presented, TAM-targeting therapies for advanced melanoma and non-melanoma skin cancer will develop in the future.

## 1. Introduction

Immune checkpoint inhibitors (ICIs), such as anti-programmed cell death 1 (PD1) antibodies (Abs) and anti-cytotoxic T-lymphocyte-associated protein 4 (CTLA4) Abs, have been widely administered for not only advanced melanoma [1,2], but also various non-melanoma skin cancers [3] such as advanced cutaneous squamous cell carcinoma (cSCC) [4], advanced basal cell carcinoma (BCC) [5], cutaneous T cell lymphomas (CTCLs) [6,7], and cutaneous angiosarcoma (CAS) [8]. The efficacy of ICIs is often higher for these than for melanoma [4]. On the other hand, the therapeutic effect of ICIs for melanoma is not as expected, especially in Asian populations [9]. The reasons involve the genetic background [10], but the tumor immune environment may also be substantially involved in resistance to ICIs [11].

Profiles of tumor-infiltrating leukocytes (TILs) play important roles in immunotherapy, especially with ICIs [12,13,14,15]. Indeed, the efficacy of anti-PD1 Abs depends on an increased number of CD8+ T cells in the tumor microenvironment [12,13]. For example, tumor-associated macrophages (TAMs) induce apoptosis of activated antigen-specific CD8+ T cells in B16F10 melanomas, leading to the induction of tolerance to ICIs [13]. Several reports have also suggested that the anti-tumor immune response caused by ICIs could modify TIL profiles [14,15]. Ipilimumab in combination with nivolumab selectively increases melanoma-specific T cells in the primary tumor to produce anti-melanoma effects [14]. Intra-tumoral administration of interferon beta (IFN-β) could increase the number of CD8+ T cells in melanoma, leading to anti-melanoma effects [15]. Dabrafenib plus trametinib with anti-PD1 Abs also increases CD8+ T cells, as well as CD4+ T cells and TAMs, leading to robust antitumor activity [16]. Moreover, plasminogen activating inhibitor-1 (PAI-1), which possesses various pro-tumorigenic functions in cancer progression and metastasis, decreases CD8+ TILs through TAMs at tumor sites, leading to the development of tolerance to anti-PD1 Abs in advanced melanoma patients [17]. In aggregate, to evaluate anti-tumor effects, it is also important to evaluate the immunomodulatory effects of each drug on tumor-resident cells such as TAMs, cancer-associated fibroblasts (CAFs), and regulatory T cells (Tregs).

## 2. Profiles of Tumor-Infiltrating Leukocytes Determine the Characteristic Immunological Background of Skin Cancer

### 2.1. Importance of Stromal Factors in Developing Skin Cancers

TAMs reside mostly in melanoma and non-melanoma skin cancers and maintain an immunosuppressive microenvironment through various pathways [18,19,20,21]. In addition, since TAMs express ICIs that directly suppress activated T cells [22], TAM-related factors such as soluble CD163 (sCD163) and its related chemokines (e.g., C-X-C motif chemokine ligand (CXCL)5, C-C motif chemokine (CCL)19, CCL26 etc.) could be useful biomarkers in patients treated with ICIs [23,24,25,26,27,28]. Since TAMs produce characteristic chemokines by the stimulation of specific stromal factors (e.g., periostin [POSTN], plasminogen activating inhibitor-1 (PAI-1), receptor activator of NF-κB ligand [RANKL]) [17,29,30], the correlations of chemokines derived from TAMs with stromal factors and profiles of TILs have been widely investigated in each cancer type [11].

#### 2.1.1. Tumor-Promoting Roles of POSTN in Skin Cancers

Fibroblasts stimulated by proinflammatory cytokines produce POSTN, a TGF-b-induced extracellular matrix protein [31,32]. POSTN is involved in cell survival, angiogenesis, and metastasis in skin cancers [32] and is expressed in stromal surrounding tumor cells in various skin cancers, including melanoma [24,29,33,34], cutaneous squamous cell carcinoma (cSCC) [35,36], basal cell carcinoma (BCC) [37], dermatofibrosarcoma protuberans (DFSP) [38], Merkel cell carcinoma (MCC) [39], and mycosis fungoides [40]. It could be a stimulator for TAMs in melanoma [1]. Since TAMs reside in the tumor sites in most of the skin cancers described above [18] and POSTN modulates TAM functions to induce proinflammatory cytokines and chemokines, as well as angiogenetic factors such as matrix metalloproteinase (MMP) [24,35,36,37,38,39,40], it is important to evaluate the expression levels of POSTN to predict tumor progression, and even predict the efficacy and immune-related adverse events (irAEs) in skin cancers [24,41].

In cutaneous melanoma, several reports suggested a correlation between POSTN expression levels and progression of melanoma, both in humans and in a mouse model [32,33,34,42]. For example, the increased stromal expression of periostin was significantly correlated with increased numbers of CD163+ TAMs in inflamed melanoma skin [35], suggesting that TAMs stimulated by POSTN play a significant role in the development of melanoma. In another report, immunohistochemical analysis showed that POSTN was expressed in invasive and metastatic human melanoma, which could accelerate melanoma progression in a mouse melanoma model [42]. More recently, RNA-Seq analysis showed that POSTN plays a central role in angiogenesis and anchorage-independent growth of melanoma [32]. Notably, POSTN stimulates M2 macrophages to produce various MMPs such as MMP12 [40]. In aggregate, POSTN could promote tumor growth by stimulation of TAMs to induce angiogenetic factors in melanoma.

Not only melanoma, but non-melanoma skin cancers also express POSTN in cancer stroma [35,36,37,38,39,40]. For example, cSCC, the second most common type of non-melanoma skin cancer, was shown to express POSTN in cancer fibroblasts (CAFs), which correlated with the aggressiveness of their biological nature [35]. Infiltrative type BCC highly expresses POSTN throughout the stroma compared to superficial type BCC, suggesting that POSTN plays a role in progression from the nodular to the infiltrative type of BCC [37]. POSTN is also expressed in the cancer stroma of DFSP almost exclusively in the proximity of tumor nodules and restricted to the mesenchymal side, where CD163+ TAMs and MMP-producing cells are abundant [38]. POSTN could be detected in the resident area of CD163+ TAMs in MCC, which is known to be an immunoreactive malignant skin tumor [39]. POSTN could even be a biomarker for the progression of Mycosis fungoides, the most common subset of CTCL, as it is expressed in the dermis in the early stage [40]. Notably, since POSTN stimulates TAMs to produce various chemokines and MMPs that might be correlated with tumor progression [40], POSTN can be a target for immune therapy in the future.

#### 2.1.2. PAI-1 in Skin Cancers

PAI-1 is a serine protease that plays a significant role in the development of skin cancers [42,43,44,45,46,47,48,49,50,51,52]. PAI-1 inhibits urokinase-type plasminogen activator (uPA) and tissue type plasminogen activator (tPA), leading to attenuation of plasminogen activation, which is associated with inflammatory migration and angiogenesis at tumor sites [42]. In melanoma, PAI-1-highly-expressing subtypes of melanoma tend to metastasize to the skin rather than lymph nodes, suggesting that PAI-1-expressing melanoma prefers hematogenous metastasis [43]. In another report, PAI-1 was found to induce resistance to chemotherapy in mouse B16F10 melanoma models [46]. Recent reports also focused on the cancer-promoting activity of PAI-1, including not only angiogenesis, but also its immunomodulatory effects [17,44]. Tseng et al. reported that PAI-1 facilitates programmed death ligand 1(PD-L1) endocytosis of melanoma cells to abrogate the efficacy of anti-PD-L1 antibodies (Abs) in mouse B16F10 melanoma models [44]. Moreover, PAI-1 expression in melanoma cells and baseline serum levels of PAI-1 are significantly correlated with the efficacy of anti-PD1 Abs in advanced melanoma patients [17]. Notably, PAI-1 stimulates TAMs to decrease Th1/Th2 chemokines such as CCL22 and CXCL10, leading to decreased TILs and increased CXCL5 to increase the migration of TAM precursors [17]. In addition, PAI-1 increases the expression of focal adhesion kinase on TAMs to facilitate the migration of macrophages into tumor sites in a melanoma model [48]. These reports suggested that PAI-1 inhibition might improve the anti-melanoma effects of ICIs. To prove this hypothesis, a phase II study investigating the safety of PAI-1 inhibitors in combination with nivolumab in the treatment of unresectable malignant melanoma is ongoing (jRCT2021210029) [47].

PAI-1 expression has also been investigated in other skin cancers such as BCC [42] and cSCC [49,50,51]. Serum levels of uPA and PAI-1 are lower in BCC than in cSCC and melanoma [52]. However, increased uPA expression in tumor cells and uPA receptor expression in the stroma surrounding tumor cells were detected in infiltrative type BCC [42], suggesting the importance of the uPA system for aggressive proliferation and infiltration of BCC cells [42]. Highly aggressive phenotypes of cSCC increase PAI-1 expression to facilitate epithelial invasive potential through MMP10 activity [50,51], contributing to the invasive patterns of cSCC [49].

In hematological malignancies, PAI-1 plays various roles in disease progression [53,54,55], and blocking of PAI-1 could be a therapeutic target [55]. PAI-1 was significantly increased in lymphoma patients compared to patients with non-metastatic solid organ malignancies [53]. Plasma levels of PAI-1 are significantly increased in the acute phase of Hodgkin’s lymphoma (HL) [54]. In chronic myeloid leukemia (CML), the TGF-β–PAI-1 axis was selectively increased in CML leukemic stem cells (LMCs) in the bone marrow, leading to expansion of the therapeutic options of tyrosine kinase inhibitors (TKIs) by blockade of PAI-1 [55]. Accordingly, phase II clinical trials proved the efficacy of PAI-1 inhbitors in combination with TKIs for the treatment of CML (UMIN000029196). These studies suggested the contribution of PAI-1 in the disease progression of CTCL, leading to develop the PAI-1 targeting therapy for advanced CTCL, though there have been no reports in English that investigated the expression of PAI-1 in cutaneous T cell lymphoma.

#### 2.1.3. Other Stromal Factors in Skin Cancer

##### RANKL as an Immunostimulatory Molecule for TAMs

Since stromal factors in each cancer site stimulate TAMs to produce characteristic chemokines and maintain the tumor microenvironment, understanding tumor-derived factors in each cancer species is also important [11,18,56,57]. For example, extramammary Paget’s disease (EMPD) is an adenocarcinoma of apocrine origin whose biological behavior is similar to that of breast cancer [13,58], and it possesses a substantial number of M2-polarized TAMs in the dermis adjacent to Paget cells [20]. Indeed, RANKL produced by Paget cells stimulates TAMs to produce CCL17, leading to migration of C-C chemokine receptor (CCR)4+ T cells such as regulatory T cells at a tumor site of EMPD [18,30]. In a mouse breast cancer model, blockade of the RANK/RANKL signal increased CD8+ T cells and reduced other types of immunosuppressive TILs such as TAMs and neutrophils, leading to enhancement of anti-tumor immunotherapies in breast cancer [58]. Interestingly, since the RANK/RANKL signal might also enhance the anti-tumor immune responses in patients treated with ICIs in several cancer species including melanoma [59,60], RANKL might be one of the best stromal targets in patients treated with ICIs in the future. Indeed, a multicenter, open label, phase 1B/2 clinical trial to evaluate the efficacy and safety profiles of nivolumab plus denosumab in an adjuvant setting is now ongoing in non-small cell lung carcinoma (ACTRN1261800112125). This clinical trial suggests that RANKL-targeting therapies for advanced melanoma and non-melanoma skin cancer will develop in the future.

##### PD-L1 Expression in Skin Cancer

PD-L1 is widely expressed on tumor cells and stromal cells such as TAMs, dendritic cells (DCs), and Langerhans cells (LCs) to maintain the immunosuppressive microenvironment in various skin cancers [23,61,62,63,64,65], not only melanoma [23], but also cutaneous SCC [61,62], Merkel cell carcinoma [63], EMPD [64,65], and CAS [66]. The expression of PD-L1 on melanoma cells is not only important to the maintenance of an immunosuppressive microenvironment, but it could also be an independent prognostic factor [67] and even represent a biomarker for predicting the efficacy of anti-PD1 antibodies in melanoma [68]. Moreover, PD-L1 expression in melanoma cells maintains the immunosuppressive function of M2-polarized PD1-expressing TAMs [69]. Importantly, blockade of PD-L1/PD1 signals by anti-PD1 Abs activates TAMs, leading to production of TAM-activating factors such as sCD163 and chemokines in the serum of melanoma patients [24,25,26,27,28]. Since these increased levels of TAM-related factors indicate the existence of anti-PD1 Abs in cancer stroma, these TAM-related factors could be useful biomarkers to predict anti-melanoma effects and onset of immune-related adverse events (irAEs) of anti-PD1 Abs in advanced cutaneous melanoma patients [24,25,26,27,28].

Since PD-L1 is expressed in tumor cells, as well as TAMs and antigen-presenting cells (APCs), to maintain the immunosuppressive microenvironment in cSCC [61], the anti-PD1 Ab cemiplimab is used for the treatment of locally advanced and metastatic cSCC [70,71]. The objective response rate (ORR) of cemiplimab is 44% [95% confidence interval (CI), 32–55] for locally advanced melanoma [4], whereas it is 45.2% (95% CI, 35.9% to 54.8%) for metastatic cSCC [70]. In Merkel cell carcinoma, the anti-PD-L1 Ab avelumab significantly prolonged overall survival (OS) in a PD-L1-highly-expressing cohort [12.9 months (95% CI, 8.7–29)] compared to a low-expressing cohort [7.3 months (3.4–14.0)] [71]. Moreover, tumor site PD-L1 expression in combination with PD-1-positive cells in TILS is a prognostic factor in CAS [66]. These reports suggested the usefulness of anti-PD1 Abs for PD-L1-expressing non-melanoma skin cancers, and a phase II clinical trial of nivolumab for the treatment of non-melanoma epithelial skin malignancies is ongoing (jRCT: 2031190048) [3].

##### IL-4 as a Stromal Factor of Skin Cancer

Since TAMs could be functionally reprogrammed and re-polarized by exposure to cytokines derived from cancer stroma, it is important to evaluate cytokines in cancer stroma in skin cancer [56,57]. For example, the production of IL-4 is increased in the cancer stroma of the advanced stage compared to the early stage in mycosis fungoides, leading to an increased ratio of CD163+CD206+ M2 polarized TAMs, which could be a measure of the transition to the advanced stage [40]. Notably, CD163+CD206+ TAMs produce M2 macrophage-related chemokines such as CCL22 by the stimulation of IL-4 in cancer stroma, which leads to recruitment of CCR4+ lymphoma cells to develop tumor formation in mycosis fungoides [72]. Since sCD163, one of the activation markers of CD163+ TAMs [73], is a disease progression marker of CTCL [74], evaluation of stromal factors that stimulate and repolarize CD163+ TAMs is important, and the neutralization of stromal factors could be a target for CTCL therapy [72,75].

### 2.2. Role of TAMs in Skin Cancers

#### 2.2.1. Chemokine Production of TAMs

Cancer stromal factors stimulate TAMs and subsequently produce characteristic chemokines and angiogenetic factors in each tumor site of skin cancers to maintain the immunosuppressive microenvironment [11,18]. In many skin cancers such as melanoma, cSCC, EMPD, and CTCL, TAMs produce immunosuppressive chemokines that attract other immunosuppressive cells such as myeloid-derived suppressor cells (MDSCs), Tregs, and tumor-associated neutrophils (TANs) [11]. For example, POSTN from CAFs stimulates TAMs to produce CXCL5 to enhance PD-L1 expression in TAMs, maintaining regulatory T cells in melanoma [76] (Figure 1). In addition, these immune checkpoint modulators (e.g., programmed death ligand 1 [PD-L1], B7-H3, B7-H4) expressed by TAMs [22] directly suppress activated T cells and maintained regulatory T cells in vivo [77]. Moreover, CXCL5 recruits monocytes as a precursor of TAMs, and since POSTN also stimulates TAMs to produce CCL22 attracting Th2 [40], IL-4 produced by Th2 polarizes monocytes to M2-polarized TAMs [11] (Figure 1). In aggregate, TAM-related chemokines develop the immunosuppressive microenvironment of melanoma, suggesting that TAMs could be a target for immunotherapy.

TAMs can be detected not only in cutaneous melanoma, but also in various non-melanoma skin cancers [11]. For example, CD163^+^ CD206^+^ TAMs that express immunosuppressive functional molecules including arginase 1 and PD-L1 were also detected together with Tregs in EMPD [20,30,65]. TAMs produce CCL17 by the stimulation of RANKL derived from Paget cells to recruit Tregs in the tumor microenvironment, maintaining the immunosuppressive microenvironment in EMPD [78]. TAMs also produce proinflammatory cytokines CCL19 and CCL21 by the stimulation of LL37 to recruit CCR7+ Foxp3+ Tregs in lesional skin of EMPD [79,80] (Figure 2). Substantial numbers of M2-polarized TAMs were also detected in cSCC, and depletion of TAMs suppressed tumor growth in cSCC [61,81,82]. Recently, Amôr et al. reported that CCL2 recruits monocytes, and these monocyte- derived macrophages matured and polarized into M2 macrophages in the lesional skin of cSCC [81]. In Merkel cell carcinoma, several reports suggested the significance of TAMs in the progression of tumors [82,83]. Substantial numbers of VEGF-C+ CD68+ CD163+ TAMs were distributed in the lesional skin of Merkel cell carcinoma [83], and expression of CD200 in tumor cells induces immunosuppressive CD163+ CD200R+ M2 TAMs and Foxp3+ Tregs to maintain the immunosuppressive tumor microenvironment [84].

As described above, TAMs play a significant role in the development of mycosis fungoides, the major subpopulation of CTCL, which could be an optimal model for their understanding [40]. Since the stromal factor of mycosis fungoides is different between the patch (early) stage and the tumor (advanced) stage, TAMs in each stage could be different. Indeed, TAMs in the patch stage produce proinflammatory chemokines (e.g., CXCL5, CXCL10), whereas in the tumor stage they produce Th2 chemokines (e.g., CCL17, CCL22) [40]. Notably, since CCL17 and CCL22 are ligands of CCR4, the major chemokine receptor expressed in CTCL cells, the production of these chemokines from TAMs could result in tumor formation in mycosis fungoides in the tumor stage. These findings suggest the necessity of different therapeutic approaches for the patch and tumor stages of mycosis fungoides [7].

#### 2.2.2. Angiogenetic Factors of TAMs

TAMs induce neovascularization by TAM-derived angiogenetic factors such as vascular endothelial growth factor (VEGF), platelet-derived growth factor, and transforming growth factor β, or by expressing matrix metalloproteinases (MMPs) [11,18]. In melanoma, VEGF-R1 stimulated by VEGF-A recruits TAMs to promote angiogenesis by the promotion of neovascularization [85,86]. In addition, VEGF-A derived from melanoma cells induces resistance to vemurafenib by the induction of VEGF receptor (R)-1 on BRAF inhibitor-sensitive counterparts [85]. In Merkel cell carcinoma, VEGF-C expressed by TAMs plays a crucial role in lymphangiogenesis and progression of Merkel cell carcinoma [84]. Moreover, in a mouse sarcoma model, blockade of the VEGF/VEGF receptor pathway inhibits M2 polarization in TAMs, abrogating angiogenesis by decreasing vascular density [87]. Furthermore, the inhibition of M2 polarization of TAMs decreased VEGF from TAMs, suppressing B16F10 melanoma growth in vivo [88]. These reports suggest the significance of VEGF produced by M2 macrophages in tumor progression.

TAMs also produce matrix metalloproteinases (MMPs), which play critical roles in the tissue remodeling associated with protein cleavage to modify the immune microenvironment, angiogenesis, tissue repair, local invasion, and metastasis [11,18]. As described above, the expression of PAI-1 was detected in the lesional skin of melanoma, which correlated with the efficacy of anti-PD1 Abs monotherapy in advanced melanoma patients [17]. Classically, PAI-1 is produced by endothelial cells in cancer sites. Notably, PAI-1 plays a significant role in the induction of other angiogenetic factors, including MMPs in skin lesions [87]. MMPs are among the central angiogenetic factors associated with M2-polarized TAMs in skin cancers [81,88]. For example, osteopontin signaling increased the secretion of prostaglandin E2 and MMP-9 from TAMs to promote angiogenesis and tumor progression in a melanoma model [88]. TAMs produce MMPs by the stimulation of stromal proteins in skin cancer [11]. MMP1 and MMP25 are produced by M2-polarized TAMs by the stimulation of RANKL, which might promote dermal invasion in EMPD and apocrine origin skin cancer [89]. The stimulation of POSTN augments the production of MMP1 and MMP12 from monocyte-derived immature M2 macrophages in CTCL [40] and DFSP [38]. Notably, transcriptional profiling showed that only MMP12 could be a risk factor for CTCL progression among MMPs [90]. Furthermore, inhibition of MMP9 expression decreases the tumor-promoting function of TAMs in melanoma [91]. These reports suggest that TAMs could be a target for angiogenetic molecular-targeted therapy in the future.

### 2.3. Other Stromal Cells in Skin Cancer

#### 2.3.1. Tumor-Associated Neutrophils (TANs)

Since TANs work as immunosuppressive cells through the production of TGF-β [92] and inducible nitric oxide synthase (iNOS) [93] in the tumor microenvironment, they drive tumor progression in skin cancer [92,93]. TANs lead to DNA base damage and mutation, leading to initiation of tumor growth [94]. These reports suggest that inhibiting TAN recruitment at tumor sites could enhance immunotherapy in skin cancer.

Recent reports suggested the significance of IL-17 in developing skin cancers such as cSCC and EMPD through the recruitment of TANs [80,95,96,97]. As a previous report suggested, IL-17 promotes the development of cSCC by keratinocyte proliferation through sustained activation of the TRAF4-ERK5 axis [95]. More recently, 7,12-dimethylbenz[a]anthracene (DMBA) and 12-O-tetradecanoylphorbol-13-acetate (TPA) were reported to induce the development of cSCC in the mouse by the enhancement of the IL-23/IL-17 signaling pathway [97]. This pathway may be correlated with the development of EMPD [80]. Notably, aryl hydrocarbon receptor (AhR) signals could trigger IL-23/IL-17 signals in both cSCC and EMPD [80,97]. These reports suggest that AhR ligands could induce IL-17 not only in skin inflammatory disease (e.g., psoriasis vulgaris), but also in non-melanoma skin cancers.

#### 2.3.2. Cancer-Associated Fibroblasts (CAFs)

CAFs comprise a phenotypically heterogeneous population of cells that play a significant role in the development of tumors and tumor microenvironments in various cancers, including skin cancers [98,99]. CAFs are among the main stromal cells, and, as described above, produce various tumor-promoting factors in skin tumors. In melanoma, CAFs express high levels of alpha-smooth muscle actin (a-SMA), one of the most important activation and differentiation markers, as well as other conventional fibroblast-activating markers such as fibroblast-activating protein (FAP), fibroblast-specific protein 1 (FSP1), osteonectin, desmin, platelet-derived growth factor receptor (PDGFR), POSTN, CD90, and vimentin [98]. Notably, due to CAF heterogeneities, all of these markers described above are not always expressed in CAFs. Therefore, CAFs may possess different pro-tumor functions in each skin cancer [98,99]. For example, the main population of CAFs in melanoma is composed of a-SMA+ CD90+ FAP+ fibroblasts, which are useful for predicting the efficacy of anti-PD1 Abs monotherapy in melanoma [100]. Indeed, CD90+ fibroblast and FAP+ fibroblast counts are significantly positively associated with progression-free survival (PFS) and overall survival (OS), whereas a-SMA+ fibroblast count is negatively associated with PFS and OS in melanoma patients treated with anti-PD1 Abs monotherapy [100]. CAFs induce resistance not only in immunotherapy but also to BRAF inhibitors in advanced melanoma [101,102]. Indeed, BRAF inhibitors enhance β-catenin nuclear accumulation through the induction of BRAF-CRAF heterodimerization and subsequent activation of ERK signaling in both CAFs and melanoma cells, leading to proliferation of melanoma cells [101]. Berestjuk et al. reported that extracellular matrix (ECM) derived from CAFs abrogate the anti-melanoma effects of BRAF/MEK inhibition in melanoma through discoidin domain receptors (DDR)1 and DDR2 phosphorylation [102]. These reports suggest that CAFs could be a target to overcome resistance to BRAF inhibitors.

CAFs induce pro-tumor effects not only in melanoma, but also in non-melanoma skin cancers. As described above (Section 2.1.1), POSTN was expressed in CAFs in various skin cancers (cSCC, BCC, EMPD, DFSP, Merkel cell carcinoma, and mycosis fungoides) [35,36,37,38,39,40], contributing to the development of the tumor microenvironment in each cancer. In BCC, the expressions of PDGFR and prolyl-4-hydroxylase mRNA were augmented among the CAF activation markers, leading to increased CAF-related chemokines CCL17, CCL18, CCL22, CCL15, CXCL12, and IL-6 in both the tumor site and peritumoral skin [98]. The tumor-derived exosome, shuttle miR-375, polarized CAFs into aSMA, CXCL2, and IL-1b-expressing phenotypes, promoting tumor growth by decreasing p53 pathway-related gene expression in Merkel cell carcinoma [103]. In mycosis fungoides, CAFs derived from mycosis fungoides increased the expression of GATA3 (Th2 marker), as well as TWIFT1 and TOX (also used as a biomarker gene for the progression of mycosis fungoides), in CTCL cells [104] Overall, these reports suggest the possible mechanisms of crosstalk between tumor cells and CAFs in the development of the tumor microenvironment in non-melanoma skin cancers.

## 3. TILs in the Tumor Microenvironment as a Target to Improve Immunotherapy

As described above, TAMs are functionally reprogrammed to polarized phenotypes by exposure to various factors, leading to the maintenance of the immunosuppressive tumor microenvironment together with other immunosuppressive cells such as Tregs, even affecting the prognosis of cancer-bearing hosts [11]. Indeed, PD-L1-expressing CD163+ TAMs were associated with significantly worse OS than a low density of TAMs (log rank *p* = 0.0025) [105] suggesting that TAM-related factors might be biomarkers for predicting the efficacy of ICI.

Since TAMs derived from melanoma express PD-1 [69], and PD-1 expression in TAMs induces M2 polarization [69], administration of an anti-PD1 antibody might repolarize and activate TAMs to release sCD163 in melanoma patients [24]. Indeed, serum levels of sCD163 were significantly increased in responders to nivolumab compared to non-responders in patients with cutaneous melanoma (sensitivity 84.6%, specificity 87.0%, respectively; *p* = 0.0030) [25]. Moreover, absolute serum levels of sCD163 after 6 weeks were significantly increased in patients treated with nivolumab who developed irAEs (*p* = 0.0018) [24]. These reports suggest that TAM-derived sCD163 could be a predictive biomarker of the efficacy and irAEs of anti-PD1 Abs.

TAM-related chemokines, CXCL5, CCL19, and CCL26, could be other prognostic markers for anti-PD1 Abs treatment [26,27,28]. CXCL5 is a chemokine that can recruit neutrophils, CXCR2+ myeloid-derived suppressor cells (MDSCs,) and CXCR2+ monocytes that can be precursors of TAMs. Moreover, CXCL5 increased PD-L1 expression in CAFs to induce an immunosuppressive microenvironment [76]. Furthermore, baseline serum CXCL5 [26] and increased serum levels of CCL19 [27] and CCL26 [28] were significantly associated with the efficacy of nivolumab in advanced melanoma. These data suggest that TAM-derived chemokines are useful to predict and improve the efficacy of anti-PD1 Abs in melanoma patients.

## 4. Conclusions

Since stromal factors and TILs are important for appropriate immunotherapy (Table 1), it is essential to evaluate the tumor microenvironment in each cancer type. Among TILs, TAMs and their related factors could be optimal biomarkers for immunotherapy such as anti-PD1 Ab therapy. In addition to immunomodulatory effects, TAMs also promote angiogenesis-promoting factors such as MMPs and VEGF by the stimulation of stromal factors such as PAI-1. For example, since PAI-1 is widely expressed in various skin cancers, clinical trials of the PAI-1 inhibitor TM5614 have been conducted in combination with standard therapy of each cancer [47]. According to the studies presented, TAMs and stromal factor targeting therapies for advanced melanoma and non-melanoma skin cancer can be developed in the future.

## Figures and Tables

**Figure 1 ijms-23-04044-f001:**
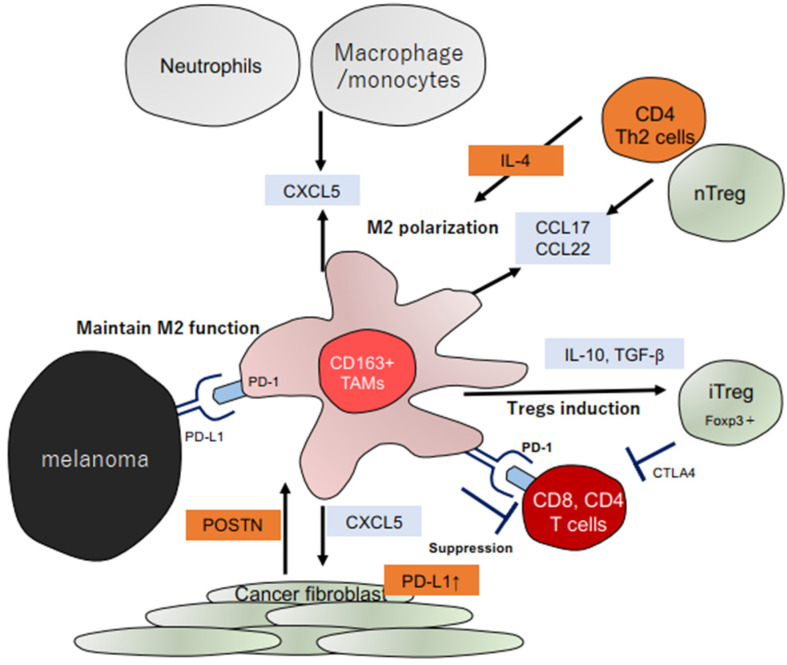
Schematic representation of TAMs in the melanoma tumor microenvironment.

**Figure 2 ijms-23-04044-f002:**
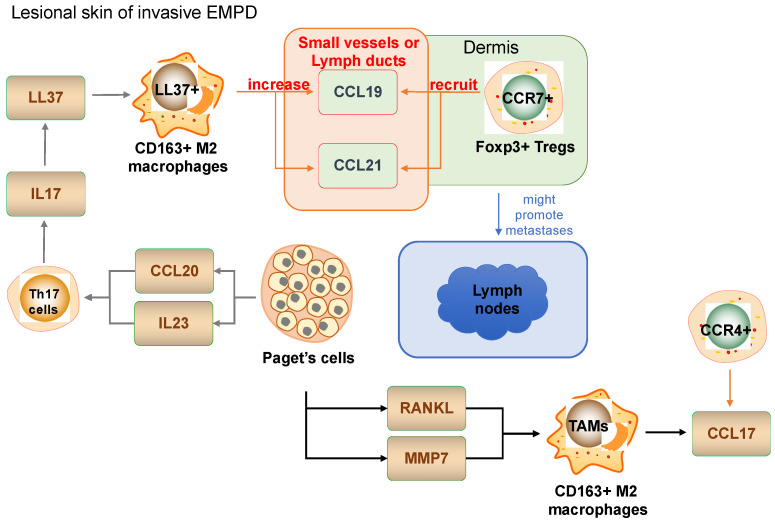
Schematic representation of TAMs in the EMPD tumor microenvironment.

**Table 1 ijms-23-04044-t001:** Possible immunological pathways in lesional skin of invasive EMPD.

	Producing Cells	Target Cells	Cancer Species	Function
POSTN	CAFs	TAMs	Melanoma	Immunomodulation
	T cells	Mycosis fungoides	Angiogenesis-promoting
	CAFs	Merkel cell carcinoma	Inflammation
	Dendritic cells		fibrosis
PAI-1	Tumor cells	Tumor cells	Melanoma	Immunomodulation
Endothelial cells	Endothelial cells	BCC	Angiogenesis-promoting
macrophages	TAMs	cSCC	Thrombus formation
adipocytes	TILs		
RANKL	Tumor cells	TAMs	Extramammary Paget’s disease	Immunomodulation
T cells	Langerhans cells	Melanoma	Osteoclast
Epithelial cells	Dendritic cells	Apocrine carcinoma	Angiogenesis-promoting
PD-L1	Tumor cells	effector T cells	Melanoma	
T cells	TAMs	cSCC	
macrophages		BCC	Immunosuppression
dendritic cells		Merkel cell carcinoma	Immunomodulation
		Other non-melanoma skin cancer	
		Cutaneous angiosarcoma

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
