# Peer review of "Stromal Factors as a Target for Immunotherapy in Melanoma and Non-Melanoma Skin Cancers"

_ijms, 2022, doi:10.3390/ijms23074044_

Round 1

Reviewer 1 Report

The manuscript entitled “Stromal factors as a target for immunotherapy in melanoma and non-melanoma skin cancers” is lacks the novelty and needs major improvements. The review written as descriptive way, like a book chapter. It should cover with more research findings and role of stomal factors which are already reported by numerous reports.

Write the mechanisms of stromal factors as targeting approach in the cancer treatment.

As per the title, the review not dealt with non-melanoma skin cancers.

The review lacks the recent updated or novel nano delivery approaches for treating melanoma, using stromal receptors as targeting.

The review must focus on patents, clinical trials, limitations, and expert opinion on the stromal receptor targeting approach for cancer treatment.

Author Response

Rebuttal comment

I cannot agree reviewer 1’s comment that my review has no novel finding. Since the purpose of review article is not for manuscript that describes a novel, unpublished finding of research but summarizes a published study, the novel finding might be limited from this point of view. Nevertheless, I even described unpublished clinical studies that still on going. I especially focused on the stromal factors that stimulate tumor-associated macrophages, and review articles of this field is still limited. For example, is there any review paper that focused on PAI-1 and TAMs?

In this review article, I focused on not only melanoma, but also non-melanoma skin cancer including cSCC, BCC, DFSP, Merkel cell carcinoma, extramammary Paget’s disease, angiosarcoma, CTCL, etc. At least, these skin cancers are non-melanoma skin cancer. Indeed, figure 2 is a scheme of tumor microenvironment of EMPD, not melanoma.

I agree that nano delivery system targeting stromal factor might develop in future, but this system for skin cancer is very limited in the field presented in this review. Since I mainly focused on immunomodulatory effects of stromal factors on immunotherapy in this review, such review article should be published independently to this manuscript.

Reviewer 2 Report

Dear authors

In the manuscript a review on the role of stromal cells in the immunotherapy of melanoma and non-melanoma skin cancer is provided.

The review is of ineptest and follow a previous review from the author. The review is well done with many details.

There are some pints that could be improved.

A table that summarized the important stromal factors and the producing cells, the function is important to have.

In the conclusions more perspectives from the authors are recommended.

Author Response

Reviewer 2

In the manuscript a review on the role of stromal cells in the immunotherapy of melanoma and non-melanoma skin cancer is provided.

The review is of ineptest and follow a previous review from the author. The review is well done with many details.

There are some pints that could be improved.

A table that summarized the important stromal factors and the producing cells, the function is important to have.

Thank you for your comment. I added the table 1 as you pointed out.

In the conclusions more perspectives from the authors are recommended.

Thank you for your comment. I have added the sentences as you pointed out.

Round 2

Reviewer 1 Report

I still not agree with the author responses. 

Author Response

I followed revised manuscript as reviewer 2 and editor suggested, and I sent you a rebuttal letter as revise 1 comment. Since there is no further comment or query from you, I did not change my revised manuscript.
